# SPA: Shape-Prior Variational Autoencoders for Unsupervised Brain Pathology Segmentation

**Cosmin I. Bercea**[2,3]                                 COSMIN.BERCEA@HELMHOLTZ-MUENCHEN.DE

**Benedikt Wiestler**[4]                                        BENEDIKT.WIESTLER@TUM.DE

**Daniel Rueckert**[3,4,5]                                       DANIEL.RUECKERT@TUM.DE

**Shadi Albarqouni**[1,2,3*]                                   SHADI.ALBARQOUNI@UKBONN.DE

[1] *Clinic for Diagnostic and Interventional Radiology, University Hospital Bonn, Germany*

[2] *Helmholtz AI, Helmholtz Zentrum Muenchen, Neuherberg, Germany*

[3] *Technical University of Munich, Munich, Germany*

[4] *Klinikum Rechts der Isar, Munich, Germany*

[5] *Imperial College London, United Kingdom*

**Editors:** Under Review for MIDL 2022

## Abstract

Deep unsupervised representation learning for brain pathology segmentation is of great interest for medical imaging, as it does not require extensive annotations for training and allows the detection of unseen pathologies. While recent approaches proposed to model the distribution of healthy brain Magnetic Resonance Imaging (MRI) using variational autoencoders, we propose to model the pixel distribution of the healthy brain by introducing a shape-prior based on the brain tissue distribution. To this end, we propose Shape-Prior variational Autoencoders (SPA) to disentangle the generative factors of brain MRI, namely cerebrospinal fluid (CSF), gray matter (GM), and white matter (WM). Our method obtains interpretable latent representations, providing pixel-wise tissue probability maps. We evaluated the proposed method on MRIs of 538 patients from six data-sets containing demyelinating lesions, small vessel disease, and tumors. Experimental results indicate that our method is capable of disentangling the generative brain MR factors and avoiding the reconstruction of anomalous regions, leading to better lesion detection performance.

**Keywords:** shape-prior, variational autoencoders, disentanglement, generative factors, brain anomaly detection

## 1. Introduction

The use of artificial intelligence for rapid and automated segmentation of abnormal structures is an important part of radiological assessment, improving clinical workflow and reducing the burden on radiologists (Hosny et al., 2018). Recently, many machine learning algorithms for identifying and segmenting critical MR findings of the brain have been proposed. Although supervised methods have been successfully applied to biomedical imaging (Ronneberger et al., 2015) and specifically to brain pathology segmentation (Baur et al., 2021b; Kamnitsas et al., 2016; Myronenko, 2019; Isensee et al., 2019; McKinley et al., 2019),

---

[*] Corresponding Author

their performance is limited by the amount and variety of annotated training data, which are sparse and expensive to collect. Furthermore, their performance in detecting abnormalities is constrained to the lesions present in the training dataset (Kamnitsas et al., 2017). To alleviate these issues, unsupervised lesion detection methods have been proposed. The underlying idea is that brain pathology can be identified as a deviation of the healthy brain distribution. As an early attempt, (Leemput et al., 2001) utilized registration to a healthy brain atlas and modeled the pixel intensities using tissue-specific probabilistic priors to detect lesions. Based on a similar concept, (Moon et al., 2002) used an atlas prior with spatial features, and (Marcel et al., 2004) combined spatial and intensity based atlas priors. (Aït-Ali et al., 2005; Freifeld et al., 2007) use probabilistic Gaussian Mixture Models (GMM) to detect and segment multiple sclerosis (MS) lesions. More recently, convolutional autoencoders are able to extract more powerful representations and are used to compress and encode healthy brain scans, then learn how to reconstruct the data back as close to the original input as possible. This allows localization and segmentation of pathology from faulty reconstructions of abnormal samples (Schlegl et al., 2017; Chen and Konukoglu, 2018; Pawlowski et al., 2018; Baur et al., 2021a; Bercea et al., 2021a; Silva-Rodrguez et al., 2021). However, autoencoders are prone to over-fitting (Steck, 2020). Variational autoencoders (VAEs) (Kingma and Welling, 2014) overcome this issue by approximating the distribution of high dimensional data and were successfully applied to brain anomaly detection (Zimmerer et al., 2018, 2019; Pinaya et al., 2021). Gaussian mixture variational autoencoders allow more complex representation in the latent space and extend VAEs to replace the uni-model Gaussian prior with a mixture of Gaussians (Dilokthanakul et al., 2017). The closest to our proposed method is the work by (You et al., 2019), who applied GMVAEs to unsupervised brain anomaly detection. In this work, we introduce novel Shape-Prior variational Autoencoders (SPA) for unsupervised brain pathology segmentation, see Figure 1. While the GMMs capture the variation in the healthy brain manifold, we propose to combine the traditional efforts in modeling the pixel distributions with the representation power and generalization capability of VAEs. To realize this, we build a probabilistic shape prior to enforce the latent disentanglement of the brain MR generative factors, namely cerebrospinal fluid (CSF), gray matter (GM), and white matter (WM). At inference, our method obtains interpretable latent representations, providing pixel-wise tissue probability maps. These probabilistic maps are then decoded to pseudo-healthy reconstructions of pathological samples, which when subtracted from the original input, results in the localization and segmentation of anomalous regions. We validate the proposed method on MR scans of 538 patients from six data-sets containing demyelinating lesions, small vessel disease, and tumors and show superior performance. Additionally, we provide insight into the latent tissue disentanglement of healthy and pathological data.

## 2. Methods

### 2.1. SPA: Shape-Prior variational Autoencoders

In contrast to previous works, we combine traditional modeling of pixel distributions with tissue-specific probabilistic brain priors with the representation power of modern VAEs. We formulate the latent variable model as $\log p(X) = \log \int p(X|Z)p(Z)dZ$, with $X \in \mathbb{R}^{H \times W}$ the image, $Z \in \mathbb{R}^{H \times W \times K}$ the latent variable and $p(Z)$ the tissue-specific probabilistic prior.

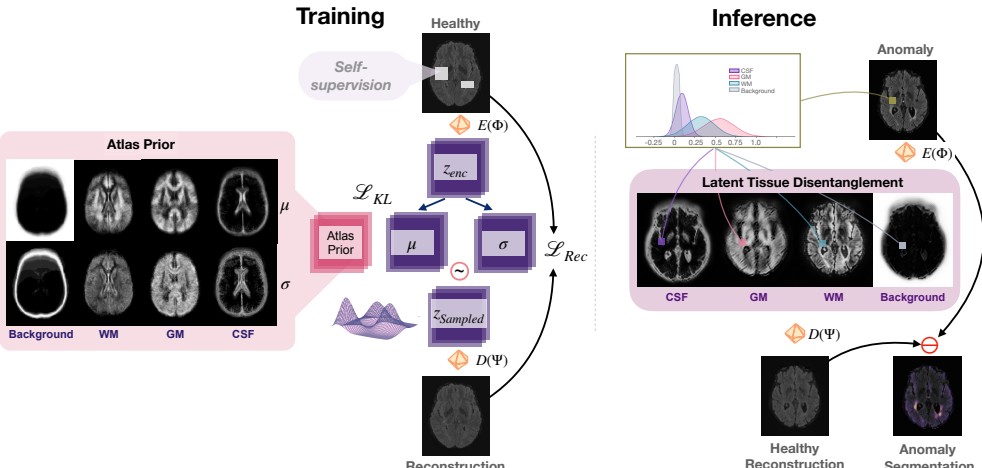

Figure 1: The proposed novel Shape-Prior variational Autoencoder disentangles the pixel-wise latent generative factors of brain MR, namely cerebrospinal fluid (CSF), gray matter (GM), and white matter (WM). The anomaly segmentation is given by the residual of the input image and its pseudo-healthy reconstruction.

To this end, we introduce the shape-prior $p(Z)$ based on the tissue distribution of healthy patients. We model every pixel $i$ in $p(Z)$ as a Gaussian-mixture model given by:

$$p(Z_i) = \sum_{c=1}^{K} \pi_{c,i} \cdot \mathcal{N}(\mu_{c,i}, \sigma_{c,i}) \tag{1}$$

with $K$ being the number of components of the mixture model: here we use $K = 4$ clusters to explicitly model the tissue channels given by CSF, GM, WM, and background; and $\mathcal{N}(\mu_{c,i}, \sigma_{c,i})$ represents the distribution of the probability of pixel $i$ belonging to cluster $c$ of a healthy population. Our objective is to maximize the log-likelihood, $\log p(X)$ of the observed samples. As the integral is intractable, we approximate the true posterior $p(Z|X)$ with a proposal distribution $q(Z|X)$ as introduced in (Kingma and Welling, 2014).

$$\log p(X) \geq ELBO = \mathbb{E}_{q(Z|X)}[\log p(X|Z)] - KL[q(Z|X)||p(Z)], \tag{2}$$

where KL denotes the Kullback-Leibler divergence; the encoder $q(Z|X)$, projects the input $X$ to the latent space $Z$ and is parameterized by the neural network $E(\Phi)$; the decoder $p(X|Z)$, reconstructs the $\hat{X}$ from $Z$ and it is parameterized by the neural network $D(\Psi)$.

## 2.2. Disentangled Representations

Disentangled learning of the latent representation are achieved by weighting the KL divergence loss term in Equation 2 with a $\beta$ term (Higgins et al., 2017). Thus, our objective is to optimize the following loss term:

$$\arg\min_{\Phi\Psi} \sum_{X} \mathcal{L}_{Rec}(X, \hat{X}) + \beta \mathcal{L}_{KL}(E(X; \Phi), p(Z)), \tag{3}$$

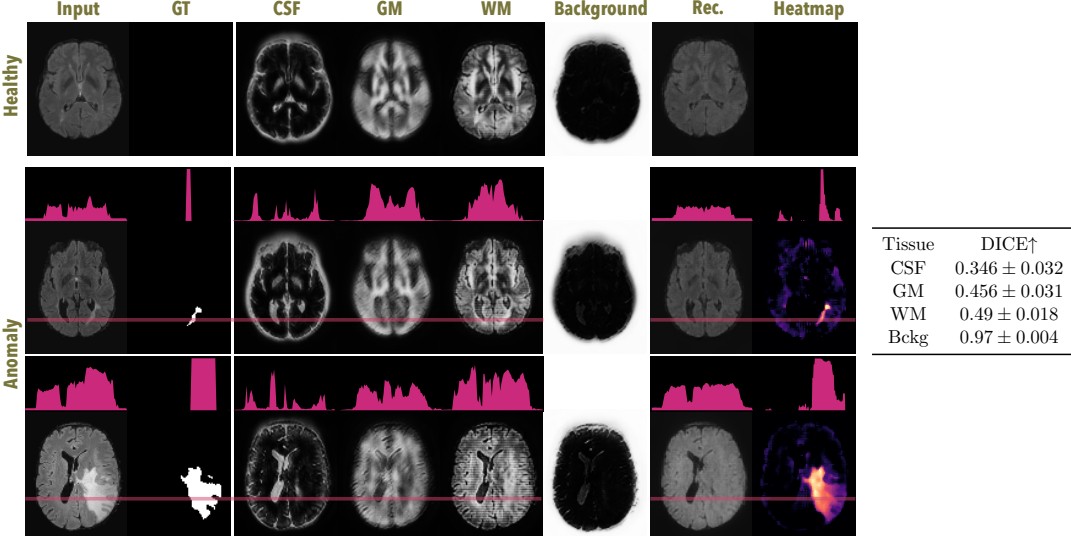

Figure 2: Visualization of the latent tissue disentanglement. The first row shows the results on healthy data, and the next two rows show tissue probability maps in presence of anomalies. Line integrals (magenta) visualize the intensity of the pixels along the line.

where $\mathcal{L}_{Rec} = \frac{1}{H \times W} \sum_{i=0}^{H \times W} |X - \hat{X}| + TV(\hat{X}) + \lambda PL(X, \hat{X})$, $\beta$ is used to weight the influence of the prior, and $\mathcal{L}_{KL}$ is the KL divergence. To enforce high-resolution and smooth reconstructions, we follow prior work for style transfer and super-resolution (Johnson et al., 2016) and add a perceptual loss and total variation regularizer to the usual mean absolute error. The main benefit of using the additional loss terms is the increased reconstruction accuracy, i.e., SSIM on healthy test data of 0.954 compared to 0.807 in our previous work (Bercea et al., 2021b). We use the same loss function for all the baselines to ensure a fair comparison. Finding the right $\beta$ term in practice is tedious. Due to the strong influence of the prior distribution, the resulting latent representations are often oversimplified and do not adequately reflect the underlying structure of the data (Dilokthanakul et al., 2017). As we show in section 4 this problem also occurs in our method and degenerates the latent tissue disentanglement. Current solutions require exhaustive search either for an annealing parameter to slowly incorporate the KL term during training (Sønderby et al., 2016), or for a cutoff threshold to stop optimizing the KL term (Kingma et al., 2017). Inspired by the signal processing literature, we propose a heuristic based on a given signal-to-noise ratio (SNR) to achieve both good reconstruction fidelity and latent regularization. Specifically, we adapt the reparameterization trick from $\mu + \sigma \odot \eta$ to:

$$\mu + \frac{1}{\sqrt{SNR}} \cdot \frac{\|\mu\|}{\|\eta\|} \cdot \sigma \odot \eta$$

where $\odot$ is the Hadamard product, $\| \cdot \|$ is the $\ell_2$-norm, and $\eta \sim \mathcal{N}(0, I)$.

Table 1: Datasets details

| | HI | MSLUB | MSISBI | MSI | WMH | GBI | BRATS |
|---|---|---|---|---|---|---|---|
| Train/Val/Test | 125/15/23 | -/-/30 | -/-/21 | -/-/48 | -/-/60 | -/-/94 | -/-/285 |
| Cohort | Healthy | | Multiple Sclerosis | | Vascular lesions | Brain tumors | |
| Age | $69 \pm 4$ | $39 \pm 10$ | N/A | $32 \pm 10$ | N/A | $61 \pm 20$ | $60 \pm 9$ |
| Sex (% female) | 56 | 76 | N/A | 61 | N/A | 50 | N/A |
| Scanner (3T) | Philips Achieva | Siemens Intera | Philips Achieva | Philips Achieva | Philips& Siemens &GE Diverse | Siemens Verio | Multiple (N=19) |
| Res ($mm$) | $1.5 \times 0.9 \times 0.9$ | $0.8 \times 0.47 \times 0.47$ | $2.2 \times 0.82 \times 0.82$ | $1.5 \times 0.9 \times 0.9$ | Diverse | $1.5 \times 0.9 \times 0.9$ | $1 \times 1 \times 1$ |
| TR/TE/TI (s) | 10/0.14/2.75 | 5/0.392/1.8 | 11/0.068/2.8 | 10/0.14/2.75 | Diverse | 5/0.395/1.8 | Diverse |

All images were re-sampled to $1 \times 1 \times 1$ mm, skull-stripped, and registered to the SRI-24 atlas (Rohlfing et al., 2010)

## 3. Experimental setup

**Datasets, Implementation and Hyperparameters.** Details of the used datasets are shown in section 3. The healthy (HI), MS lesions (MSI) and glioblastoma (GBI) internal databases are provided by Klinikum Rechts der Isar in Munich. We use public datasets of MSLUB (Lesjak et al., 2018), MSISBI (Carass et al., 2017), WMH(Kuijf et al., 2019), and BraTS (Menze et al., 2015; Bakas et al., 2017, 2019). Both the common encoder $E(\Phi)$ and decoder $D(\Psi)$ follow the original U-Net description by Ronneberger et al. (Ronneberger et al., 2015), but adjusted the network to handle input at a $128 \times 128$ resolution. The resulting network has 3 blocks each for encoding and decoding. The composition of a block is as follows: $2 \times [3 \times 3$ convolutions with filters $\in \{32, 64, 128\}$ , batch normalization and LeakyReLu activation]. The mean and log sigma estimation of VAEs is implemented using two convolution only layers with a kernel size of 1. For the implementation of SPA, we additionally add a *sigmoid* and *logsigmoid* activation layers to bring the estimated distribution in the range of the prior. We follow (You et al., 2019) for the GMVAE latent implementation. We derived the distribution of the tissue specific probabilistic prior on 136 healthy patients from our healthy internal dataset HI. We calculated the mean and variance of the atlas distribution by averaging segmentation maps of CSF, GM, WM and background of 136 healthy patients, that were obtained using the antsAtroposN4.sh script (Avants et al., 2009). Note that, our method is not dependent on the derived atlas and can be extended with any distribution over the tissue-specific probability maps. For all baselines, we used self-supervision techniques as described in (Bercea et al., 2021b) to enforce healthy reconstructions and trained the networks until convergence with a batch size of 2, and ADAM optimizer with a learning rate of $5 \cdot 10^{-4}$ and exponential decay of 0.97. We set the $\lambda := 0.1, \beta := 10^{-3}$, and $SNR := 16$ as discussed in section 4.

**Pre- and post-processing.** Similar to (Baur et al., 2021a), all scans have been registered to the SRI24 atlas template space (Rohlfing et al., 2010) and have subsequently been skull-stripped with ROBEX (Iglesias et al., 2011) and normalized to the [0,1] range. We used all axial slices with visible tissue information with a size of 128×128px for training and evaluation. We keep the absolute residual of the input and its reconstruction. We use the generated heat maps to compute AUPRC and binarize the results using a threshold $\tau$ to compute the maximum DICE scores per dataset.

**Evaluation metrics.** We observe both L1 and KL losses on validation data for the model selection. To measure the anomaly segmentation performance and compare different models,

Table 2: We show AUPRC to assess anomaly detection on six datasets with pathology. We compare our method to sota unsupervised models without any post-processing (Baur et al., 2021a), namely: context VAE (c-VAE) (Zimmerer et al., 2018), constrained adversarial AEs (cAAE) (Chen and Konukoglu, 2018), f-AnoGAN (Schlegl et al., 2017), and Gaussian-mixture VAEs with image restoration (GM-IR) (You et al., 2019); two variational baselines with the same training and implementation details as our network; and two supervised methods based on U-Net (Ronneberger et al., 2015). (*) shows results with simple post-processing (keeping positive residual and applying median filtering of size 3); and (•) shows results on a validation subset; **bold**/underscore show the best two results among baselines.

| Training Method | Sota Unsupervised - AUPRC ↑ | | | | | | | | AUPRC ↑ | | |
| | Healthy | | | | | | | | Supervised | | Thresh. |
| | c-VAE | cAAE | f-AnoGAN | GM-IR | VAE | GMVAE | SPA | SPA* | MSLUB | BraTS | n/a |
|---|---|---|---|---|---|---|---|---|---|---|---|
| MSLUB | 0.030 | 0.035 | 0.034 | 0.037 | 0.059 | **0.078** | 0.059 | 0.132 | 0.420• | 0.049 | 0.150 |
| MSISBI | 0.023 | 0.027 | **0.035** | 0.025 | 0.007 | 0.011 | 0.023 | 0.019 | 0.423 | 0.020 | 0.135 |
| MSI | 0.027 | 0.036 | 0.041 | 0.036 | 0.092 | 0.162 | **0.178** | 0.389 | 0.478 | 0.094 | 0.114 |
| WMH | | n/a | | | 0.027 | 0.101 | **0.159** | 0.167 | 0.407 | 0.147 | 0.280 |
| GBI | 0.052 | 0.076 | 0.061 | 0.086 | 0.160 | 0.209 | **0.234** | 0.453 | 0.267 | 0.463 | 0.367 |
| BraTS | | n/a | | | 0.104 | 0.215 | **0.320** | 0.397 | 0.350 | 0.751• | 0.473 |

we report the area under the precision $TP/(TP + FP)$ - recall $TP/(TP + FN)$ curves (AUPRC), with $TP$, $FP$, and $FN$ being true positives, false positives and false negatives.

## 4. Results

We experimentally validated our method on multiple institutes with real pathological datasets containing MS-, vascular-, and brain tumor lesions. Our main findings are i) we successfully disentangle the generative factors of brain MR imaging, namely, the CSF, GM, WM, and background; ii) we avoid the reconstruction of unwanted pathological features; and consequently iii) we improve the anomaly detection performance.

**Latent Tissue Disentanglement.** Our proposed method aims at disentangling the generative factors of brain MR and provide latent pixel-wise tissue probability maps. Figure 2 provide quantitative and qualitative results of the learned latent representations. We show the Dice score of predicted tissue maps on 30 healthy unseen test subjects. Visually, the tissue segmentation maps capture the underlying anatomy and intracranial tissue distribution despite the lower Dice scores. In particular for CSF one can observe a tendency to also segment CSF surrounding the brain, where the skull would normally be. However, considering that a small rim of (sub)dural CSF surrounds the brain and the space surrounding it has a very similar (low) intensity as CSF on these FLAIR images as a result of the skull stripping, the segmentation makes sense. The first row in the figure visualizes the tissue probability maps on a healthy patient, followed by two examples containing pathology. The provided intensity profiles show the intensity of the pixels along the marked line. Note that, while anomalies correspond to a peak in the intensity profile of the input, finding a common threshold for both cases is not straightforward. For example, the peak in the first row would correspond to the intensity of healthy tissue of the second case. Interestingly,

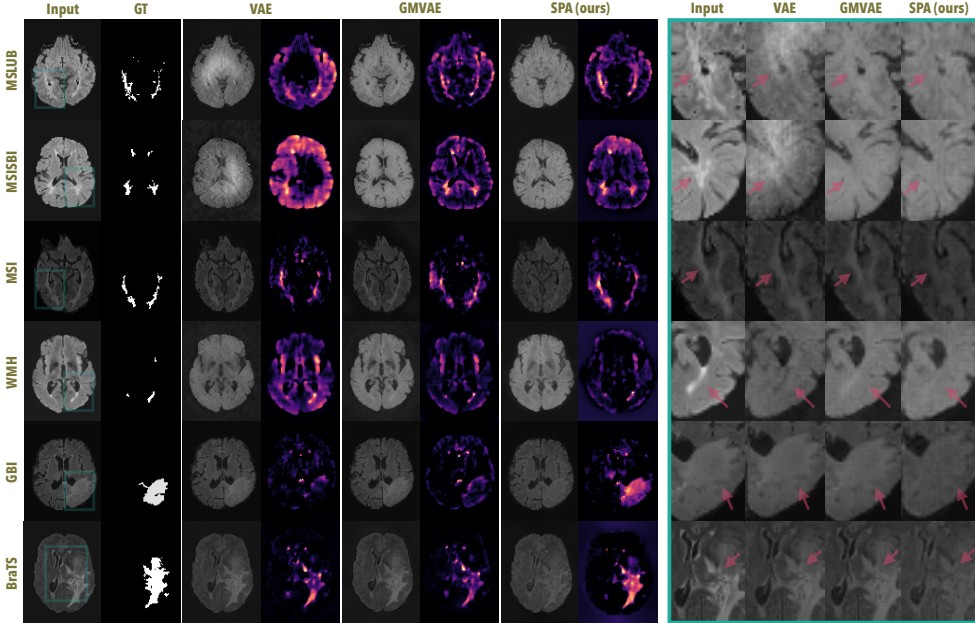

Figure 3: Qualitative results of our method compared to the VAE and GMVAE baselines. The first columns show the input and ground truth segmentation mask followed by reconstructions and residual heatmaps for the different datasets (rows). The last four columns show a zoomed-in image highlighting the reconstruction of unwanted pathological features.

the anomalous regions are either encoded as WM or GM in the latent tissue maps, having the same probability as healthy tissues. The decoder will thus only reconstruct intensities of healthy tissues containing CSF, GM, and WM, as can be seen in the line profiles. Finally, note that due to the pseudo-healthy reconstructions, the residual maps and their line profiles are very similar to the ground truth, making anomalies easy to detect.

**Anomaly Segmentation.** Table 2 shows the anomaly segmentation performance of our proposed method. We outperform sota unsupervised anomaly detection methods without any post-processing, see Table 8 in (Baur et al., 2021a) and achieve an average improvement (AUPRC) of 154.5%, 177.5% and 35.5% over the best unsupervised method, GM-IR, and variational baselines VAE and GMVAE, respectively. Compared to supervised methods, our proposed method outperforms or achieves competitive anomaly results with different U-Net variants. Even though not directly comparable, a supervised U-Net validated on the dataset it was trained on achieves the best results. This indicates that our method generalizes better to unseen domains, and that the performance of supervised method is degraded on pathologies not seen during training, reconfirming the results in (Baur et al., 2021b). The commonly used approach in clinical practice is a naive threshold-based classifier that selects hyper-intense regions. With just a simple post-processing step (keeping positive residuals and applying a median filter of size 3) our method achieves remarkable relative DICE improvement on MSLUB (24%), MSI (293%), and GBI (16%) and slightly worse on MSISBI, WMH and BraTS, reconfirming the findings of (Baur et al., 2021a) that tumors might be quite challenging since it might contain lesions that do not appear as hyperintensity

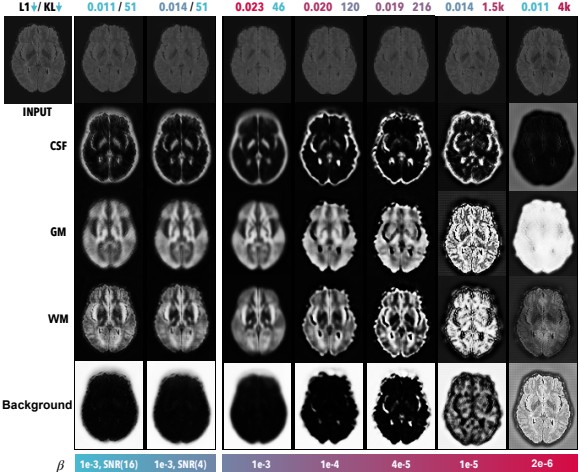

| Method $\rightarrow$ | Baselines | | | | Ours | |
| | AE | | GMVAE | | PMVAE | |
| $\beta \downarrow$ | L1 | KL | L1 | KL | L1 | KL |
|---|---|---|---|---|---|---|
| $1e^{-3}$, S16 | 0.009 | 1 | 0.007 | 1 | 0.011 | 51 |
| $1e^{-3}$, S4 | 0.015 | 1 | 0.009 | 1 | 0.014 | 51 |
| $1e^{-3}$ | 0.023 | 16 | 0.028 | 13 | 0.023 | 46 |
| $1e^{-4}$ | 0.020 | 120 | 0.023 | 126 | 0.020 | 120 |
| $4e^{-5}$ | 0.018 | 320 | 0.020 | 350 | 0.019 | 216 |
| $1e^{-5}$ | 0.012 | 2k | 0.013 | 2k | 0.014 | 1.5k |
| $2e^{-6}$ | 0.008 | 7k | 0.010 | 7k | 0.011 | 4k |

Figure 4: Model Selection. We show quantitative and qualitative results on the healthy validation set for multiple choices of $\beta$. We set $\beta$ to $1e^{-3}$ and SNR to 16 for our experiments.

in FLAIR, and naive thresholding should be still considered as a strong baseline(Meissen et al., 2021). Figure 3 gives more insight into the network's predictions. Our method avoids the reconstruction of pathology (best seen in the last columns) and delivers more accurate segmentation masks.

**Sensitivity to $\beta$ values.** Figure 4 shows quantitative and qualitative results of the effect of $\beta$. Note that, high values of $\beta$ enforce the posterior to match the prior atlas distribution, however at the cost of the reconstruction accuracy. Setting $\beta$ to lower values yields better reconstructions, but the posterior diverges from the prior atlas distribution. Finding the balance between the two is a tedious task and leads in most cases to either a compromise on the reconstruction or regularization. The $\beta$ values of $10^{-3}$ and SNR of 16 achieve the best validation results in terms of reconstruction loss and KL divergence for all methods.

## 5. Discussion and future work

In this paper, we proposed Shape-Prior variational AEs (SPA), to combine traditional modeling of pixel distribution of brain MR based on tissue probabilistic priors and the representation and generalizability of VAEs. We generate the probabilistic shape prior once, offline, from affine-registered healthy scans to avoid the requirement of good alignment of input scans during inference. We have showed that our method provides interpretable pixel-level latent tissue segmentation maps, enforces the anomalous pixels to be mapped to the healthy pixel-wise distribution and considerably improves the results without post-processing over sota unsupervised methods. However, post-processing steps are still critical for better results and their effects should be studied in the future. We find that our method performs best on datasets that follow the same distribution as the healthy training set and plan to investigate and address the domain shift in data from multiple institutes and its impact on generalization with the help of disentanglement and federated learning.

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

## Appendix A. Quantitative Metrics

Table 3 shows area under precision-recall curve for supervised U-Net (Ronneberger et al., 2015) trained and evaluated on different datasets. Table 4 shows the best achievable DICE and mean hausdorff distance (HD). Table 5 shows results with a simple post-processing step, where we keep the positive residual, instead of the absolute error since pathology tends to appear as hyper-intense regions in FLAIR sequences and apply median filtering of size 3 to the result.

Table 3: We show AUPRC to assess the anomaly detection performance on six different datasets with pathology for supervised methods based on U-Net (Ronneberger et al., 2015)

| Method → | AUPRC ↑ | | | | | |
|---|---|---|---|---|---|---|
| Trained on | Supervised | | | | | |
| Task ↓ | U-Net(Ronneberger et al., 2015) | | | | | |
| | MSLUB | MSISBI | MSKRI | WMH | GBKRI | BraTS |
| MSLUB | 0.420• | 0.423 | **0.478** | 0.407 | 0.267 | 0.350 |
| MSISBI | 0.262 | 0.536• | 0.162 | **0.480** | 0.136 | 0.156 |
| MSI | 0.328 | 0.158 | 0.656• | **0.466** | 0.174 | 0.176 |
| WMH | 0.234 | **0.501** | 0.135 | 0.516• | 0.145 | 0.191 |
| GBI | 0.102 | 0.015 | 0.264 | 0.060 | 0.551• | **0.454** |
| BraTS | 0.049 | 0.020 | 0.094 | 0.147 | **0.463** | 0.751• |

Table 4: We show the maximum DICE scores per dataset given by $2TP/(2TP+FP+FN)$ and the mean hausdorff distance $\lceil HD \rceil$ to assess the anomaly detection performance on six datasets with pathology.

| Dataset | $\lceil Dice \rceil$ ↑ | | | | $\lceil HD \rceil$ ↓ | | | |
|---|---|---|---|---|---|---|---|---|
| | VAE | GMVAE | SPA(Ours) | Thresh. | VAE | GMVAE | SPA (Ours) | Thresh. |
| MSLUB | 0.133 | **0.185** | 0.136 | 0.219 | **30.2** | 32.1 | 31.2 | 24.0 |
| MSISBI | 0.022 | 0.029 | **0.055** | 0.208 | 49.9 | 49.4 | **37.6** | 24.1 |
| MSI | 0.210 | **0.306** | 0.275 | 0.157 | **29.4** | 32.3 | 35.3 | 23.7 |
| WMH | 0.073 | 0.204 | **0.241** | 0.327 | 34.6 | 33.0 | 36.3 | 25.3 |
| GBI | 0.260 | 0.281 | **0.313** | 0.386 | 39.1 | **38.4** | 44.3 | 24.3 |
| BraTS | 0.209 | 0.297 | **0.388** | 0.443 | 37.7 | **35.3** | 57.3 | 21.6 |

Table 5: Results with simple post-processing (keeping just positive residuals and applying median filtering of size 3). We show AUPRC, the maximum DICE scores per dataset given by $2TP/(2TP + FP + FN)$ and the mean hausdorff distance $\lceil HD \rceil$ to assess the anomaly detection performance on six datasets with pathology.

| | | AUPRC ↑ | | | |
|---|---|---|---|---|---|
| Dataset | VAE | GMVAE | SPA(Ours) | Thresh. |
| MSLUB | 0.051 | 0.128 | **0.132** | 0.135 |
| MSISBI | 0.006 | 0.015 | **0.019** | 0.119 |
| MSI | 0.162 | 0.312 | **0.389** | 0.091 |
| WMH | 0.087 | 0.098 | **0.167** | 0.278 |
| GBI | 0.165 | 0.384 | **0.453** | 0.432 |
| BraTS | 0.087 | 0.324 | **0.397** | 0.487 |

| Dataset | $\lceil Dice \rceil$ ↑ | | | | $\lceil HD \rceil$ ↓ | | | |
|---|---|---|---|---|---|---|---|---|
| | VAE | GMVAE | SPA (Ours) | Thresh. | VAE | GMVAE | SPA (Ours) | Thresh. |
| MSLUB | 0.110 | 0.190 | **0.217** | 0.175 | 29.2 | **26.1** | 29.4 | 28.2 |
| MSISBI | 0.027 | 0.036 | **0.044** | 0.176 | 35.8 | **31.4** | 35.6 | 28.9 |
| MSI | 0.242 | 0.376 | **0.437** | 0.111 | **24.7** | 25.7 | 25.3 | 24.8 |
| WMH | 0.160 | 0.201 | **0.224** | 0.318 | 35.5 | **33.2** | 33.7 | 29.7 |
| GBI | 0.258 | 0.395 | **0.496** | 0.427 | 30.4 | 29.4 | **29.2** | 23.0 |
| BraTS | 0.185 | 0.415 | **0.421** | 0.446 | 54.0 | 28.6 | **26.2** | 20.6 |

