# OpenReview forum: "SPA: Shape-Prior Variational Autoencoders for Unsupervised Brain Pathology Segmentation"
_MIDL.io/2022/Conference — Submitted to MIDL 2022_

### Official Review · Reviewer_KuPN · 2022-01-18

**Confidence:** 4
**Preliminary Rating:** 4
**Recommendation:** Oral, Poster

**Summary:**

The authors propose a shape-prior variational autoencoder in order to disentangle the pixel-wise latent generative factors (three different tissue types) of brain MRIs. In their approach each pixel is modeled as a Gaussian mixture model in the latent space containing the four clusters corresponding to tissue types and background. Next to the disentanglement of the tissue types, the authors claim that their approach has a (more) interpretable latent space. The performed experiments aim for an unsupervised brain pathology segmentation by training the proposed autoencoder on healthy brain images and detecting pathological structures as reconstruction errors. The experiments for tissue disentanglement and pathology segmentation are evaluated thoroughly, both, quantitatively and qualitatively.

**Strengths:**

The idea of using a shape-prior constraint on the latent space is interesting and yields autoencoders with a more intuitive and interpretable latent spaces. The paper is overall well written and easy to follow and understand. The results are evaluated thoroughly and compared to other methods. Especially, the qualitative evaluation shows the advantages of the method as opposed to other autoencoder-based pathology detection approaches. The choice of evaluation techniques is adequate. Many different datasets including public datasets are used in the experiments.

**Weaknesses:**

- The quantitative results do not look convincing. The AUPRC and Dice values for the pathology detection are very low. On some datasets, as it seems, even supervised methods fail to produce adequate segmentation results, which may indicate that they are not particularly suitable or their ground truth segmentations are not reliable. The best results are indeed achieved on private datasets. This is surely, due to using image from the same domain for training, however this hinders comparability.
- According to the results from Fig. 2 the disentanglement of the four tissue types is not accurate. With the best Dice of 0.49 for white matter, those results are not very convincing (especially for CSF). Of course some plausible segmentation is achieved in the latent space, yet, it does not seem to accurately correspond to the tissue types.
- The interpretation of the qualitative results, e.g. Fig. 2 is misleading. The authors state that the anomalous regions are difficult to spot in the line profiles of the input image, however, in my viewing the pathological regions clearly correspond to a peak in the  presented line profiles. Furthermore, the line profiles of the reconstructed images seem more smooth in general. For example, the first image has a small "dip" on the line crossing the ventricle, which completely disappears in the reconstruction.
- The explanation of the sensitivity of $\beta$ values does not completely correspond to Fig. 4. The authors claim "Setting $\beta$ to lower values yields better reconstructions" , however, for $\beta=1e^{-3}$ and $\beta=2e^{-6}$ the reconstruction is the same. Fig. 4. needs more explanations, it is very hard to understand.
- In my understanding, the method is also only applied for 2D slices.


**Deanonymize Review:**

no

**Detailed Comments:**

Further minor problems include:
- In sec. 2.2. it is not clear how signal-to-noise (SNR) is computed. Later on it seems that SNR is chosen to be a constant.
- Some unfinished sentences/citations and grammatical issues make the work harder to follow (Page 5, citation ber and unfinished sentence "Note that, our method is not dependent ON the", Page 7, Sec. 4.2. "[...] than evaluated on.")
- The self-supervision is hardly explained.

**Paper Type:**

both

**Questions To Address In The Rebuttal:**

- Better interpretation of the qualitative results
- Explanation and clarity improvement of Fig. 4
- Have you tried training on a different healthy training dataset that might be more compatible with the other anomalous image domains?

**Special Issue:**

no

---

### Official Review · Reviewer_47K2 · 2022-01-24

**Confidence:** 4
**Preliminary Rating:** 2

**Summary:**

In this paper, the authors address the problem of unsupervised brain pathology segmentation. To tackle this, they propose a method based on a VAE that they improve by constraining the latent space with brain tissue information. Results obtained in the experimental setup show that the proposed method yields a better segmentation performance than two different VAE-based approaches.

**Strengths:**

-The paper is well written and easy to read. The methodology is sound.

-The addressed problem is interesting and relevant for the MIDL community.

-Using knowledge-driven priors has attracted quite a lot of attention recently, and using shape-priors of healthy structures in unsupervised lesion detection is interest


**Weaknesses:**

The main weaknesses I see are in the evaluation of the proposed method:
- The reconstruction loss was modified to include a perceptual loss and a total variation regularizer and the reparametrization trick formulation was also modified. The influence of those changes is not studied in the experimental setup. An ablation study to compare the performance with a model using the vanilla reconstruction loss and reparametrization trick is necessary to see if the performance improvement is coming from the disentangled VAE formulation, those formulation tweaks or a combination of both. Also, were those changes applied in the implementation of the compared methods? If not, the comparison would be unfair.
- The model evaluation is not convincing, with several recent works in the same task not included in the comparison to the proposed method (See [a-c]). Furthermore, the authors argue that the choice of included methods is based on the high performance reported in the recent survey [Baur et al,21]. However, looking at the reported tables in that survey, there exist some other methods that perform well, which are not included in the evaluation. For example, Context VAE and Bayesian VAE sometimes outperform (or at least work on par) with both f-AnoGAN and Restoration-VAE. Thus, I strongly recommend including the methods in [a-c], as well as Context VAE and Bayesian VAE in the experiments.
[a] You S, Tezcan KC, Chen X, Konukoglu E. Unsupervised lesion detection via image restoration with a normative prior.  MIDL'19
[b] Silva-Rodríguez J, Naranjo V, Dolz J. Looking at the whole picture: constrained unsupervised anomaly segmentation. BMVC'21
[c] Chen X, Konukoglu E. Unsupervised detection of lesions in brain MRI using constrained adversarial auto-encoders. arXiv preprint arXiv:1806.04972. 2018 Jun 13.
- While DSC is a coarse estimator of the quality of the segmentation, typically distance-based metrics, such as the HD are also used to report the results.

Apart from that, it seems to me that the proposed method introduces additional complexity in the processing pipeline, as it requires additional information (the CSF, GM, WM and background maps) which is obtained in the preprocessing phase. It would be good to address this additional complexity in the discussion part.

**Deanonymize Review:**

no

**Detailed Comments:**

Nothing to add.

**Final Rating After The Rebuttal:**

3: Borderline

**Justification Of The Final Rating:**

I feel the evaluation of the paper could be stronger by comparing the proposed methods to recent advances.
However, the paper is well written and overall of good quality and so I am increasing my rating to borderline.

**Paper Type:**

methodological development

**Questions To Address In The Rebuttal:**

The main interrogations concern the evaluation of the model, especially the omission of some relevant methods in the method comparison section.
Any clarification on this and on the points listed in the previous sections will be welcome and appreciated.

**Special Issue:**

no

---

### Official Review · Reviewer_5WBs · 2022-01-25

**Confidence:** 4
**Preliminary Rating:** 4
**Recommendation:** Oral, Poster

**Summary:**

This paper presents a novel approach for unsupervised brain anomaly segmentation, which applies a "shape prior" in the latent space of a Variational Autoencoder (VAE). This prior is modeled as a mixture of Gaussian distributions where each component represents a different tissue class. The proposed approach is tested on the segmentation of brains lesions in MRIs from the MSLUB, MSISBI, WMH and BraTS datasets. Results show this approach to outperform other methods based on auto-encoders (AE, VAE and GMVAE).

**Strengths:**

* Although inspired by the GMVAE approach by You et al., the proposed method also brings novel elements. Specifically, it extends GMVAE by imposing a shape prior on the latent representation, which is derived from a statistical atlas. This helps the model learn tissue-specific representations of input images, which can better identity anomalies in these images. It also increases the interpretability of predictions since the representations of different tissues are disentangled.

* The usefulness of the proposed method is demonstrated on four different datasets. Results show clear improvements compared to state-of-the-art approaches like GMVAE and several baselines.

* The paper is well written and structured.


**Weaknesses:**

* Experiments could be improved by including an ablation study.

* Some elements of the methodology, such as the shape prior and the heuristic employed in the reparameterization trick, could be better motivation and explained.

* While I understand that unsupervised segmentation is extremely challenging, given the low segmentation accuracy of the method, I am not entirely convinced that the method could be useful in clinical practice.

**Deanonymize Review:**

no

**Detailed Comments:**

* The reparameterization trick base in the signal-to-noise ratio is interesting but could be better motivated. Also, what happens if ||eta|| ~ 0 ?

* The authors replaced the standard L2 loss with a custom reconstruction loss combining three terms: L1, total variation (TV) and perceptual loss. It would be useful to include an ablation study evaluating the respective contributions of these terms.

* A U-Net is used for the encoder and decoder of the proposed model. However, the output of a standard U-Net had the same resolution and its input, not a reduced size as the latent vector. I suppose the U-Net includes both the encoder and decoder, but then wouldn't the skip connections make the reconstruction trivial?

* I am not sure how the atlas prior is computed. From Fig. 1, it looks like the prior at each pixel is a categorical distribution with a probability for each class, but Eq. (1) models it as a mixture of Gaussian. This important point should be clarified.

* It seems that using the atlas prior requires a good alignment to a template. It would be interesting to evaluate and/or discuss the sensitivity of the method to this alignment?

Other comments:

* Section 1: "to pseud-healthy reconstructions" --> "to pseudo-healthy reconstructions"

* Section 4: "we successfully disentangles" --> "we successfully disentangle"

* Section 4: "DICE" --> "Dice"

**Final Rating After The Rebuttal:**

4: Weak Accept

**Justification Of The Final Rating:**

I thank the authors for their detailed answers to my comments. Several details about the method are now clearer. While I believe the paper should be accepted, the poor clinical motivation (how such low accuracy can be translated to a useful application) refrains from giving a score of Strong Accept.

**Paper Type:**

methodological development

**Questions To Address In The Rebuttal:**

The rebuttal should clarify the points mentioned in the comments, in particular the computation of the shape prior (GMM parameters) and the choice of heuristic in the reparameterization trick. Authors should discuss the usefulness of their method in clinical practice given that the segmentation scores are low. If possible, they should also include ablation experiments or least discuss the impact of different losses on performance.

**Special Issue:**

no

---

### Meta-Review · Area_Chair_BFxm · 2022-02-19

**Recommendation:** Accept (Poster)
**Confidence:** 5

**Metareview:**

This work initially received mixed scores, with 2 weak accepts and 1 weak reject. After reading reviewers' concerns, authors rebuttal and the manuscript, I believe the paper has some merits to be presented at MIDL. Nevertheless, I side with reviewer 47K2 in that the evaluation is weak, as authors omitted recent relevant papers in their comparison. Thus, I strongly recommend the authors to include additional works suggested by reviewer (and potentially others not included in their work) in the final camera ready version, as I found the answer to this specific concern unconvincing. Other than that, I think the authors did a good job in the rebuttal and I recommend acceptance (as poster) for this work.

---

### Decision · Program_Chairs · 2022-02-28

**Decision:**

Reject

**Comment:**

Updated decision 11.04.2022
Based upon the request of the authors, this article has been withdrawn.